# Novel 3D Printed Resin Crowns for Primary Molars: In Vitro Study of Fracture Resistance, Biaxial Flexural Strength, and Dynamic Mechanical Analysis

**DOI:** 10.3390/children9101445

**Published:** 2022-09-22

**Authors:** Nayoung Kim, Hoon Kim, Ik-Hwan Kim, Jiho Lee, Ko Eun Lee, Hyo-Seol Lee, Jee-Hwan Kim, Je Seon Song, Yooseok Shin

**Affiliations:** 1Department of Pediatric Dentistry, College of Dentistry, Yonsei University, Seoul 03722, Korea; 2Research Institute of Agriculture and Life Sciences, College of Agriculture and Life Sciences, Seoul National University, Seoul 08826, Korea; 3Department of Pediatric Dentistry, Yonsei University Dental Hospital, Seoul 03722, Korea; 4Department of Mechanical Engineering, Korea Advanced Institute of Science and Technology (KAIST), Daejeon 34141, Korea; 5Department of Pediatric Dentistry, Kyung Hee University Dental Hospital, Seoul 02447, Korea; 6Department of Prosthodontics, College of Dentistry, Yonsei University, Seoul 03722, Korea; 7Department of Conservative Dentistry, College of Dentistry, Yonsei University, Seoul 03722, Korea

**Keywords:** 3D printing, mechanical properties, fracture resistance, biaxial flexural strength, dynamic mechanical analysis, primary molar, 3D printed resin crown

## Abstract

This study evaluated the fracture resistance, biaxial flexural strength (BFS), and dynamic mechanical analysis (DMA) of three-dimensional (3D) printing resins for the esthetic restoration of primary molars. Two 3D printing resins, Graphy (GP) and NextDent (NXT), and a prefabricated zirconia crown, NuSmile (NS), were tested. GP and NXT samples were 3D printed using the workflow recommended by each manufacturer. Data were collected and statistically analyzed. As a result of the fracture resistance test of 0.7-mm-thick 3D printed resin crowns with a thickness similar to that of the NS crown, there was no statistically significant difference among GP (1491.6 ± 394.6 N), NXT (1634.4 ± 289.3 N), and NS (1622.8 ± 323.9 N). The BFS of GP was higher for all thicknesses than that of NXT. Both resins showed high survival probabilities (more than 90%) when subjected to 50 and 150 MPa. Through DMA, the glass transition temperatures of GP and NXT were above 120 °C and the rheological behavior of GP and NXT according to temperature and frequency were analyzed. In conclusion, GP and NXT showed optimum strength to withstand bite forces in children, and 3D printed resin crowns could be an acceptable option for fixed prostheses of primary teeth.

## 1. Introduction

Esthetic dentistry has become an essential component of modern pediatric dentistry [1,2]. Parents’ demands for esthetic solutions when restoring their children’s teeth are increasing these days [3,4]. In addition, children themselves want dentists to restore their decayed teeth to their original appearances [5,6].

The treatment of decayed primary teeth has always been challenging for clinicians. For children who present extensive, multi-surface lesions or high caries-risk, the American Academy of Pediatric Dentistry advocates for the use of full-coverage restorations. The most frequently used restoration has been a preformed stainless-steel crown (SSC). SSC is recommended due to its long-term durability, less recurrent caries, low cost, and ease of preparation and placement [7]. Despite these benefits, parents and patients are unsatisfied with the color of SSC owing to its metallic appearance [5,8]. Various attempts have been made to overcome this esthetic problem by introducing open-faced SSC, pre-veneered SSC, and zirconia crowns. An open-faced SSC has a facial window cut, wherein the composite resin is bonded onto the tooth. A pre-veneered SSC has a composite or porcelain coating, which is mechanically and chemically bonded to a metal. Both of these crowns have superior esthetics compared to conventional SSC; however, they also have several disadvantages, including the need for more preparation, inability to bend the edges, high costs, and the tendency of esthetic coatings to be fractured [9,10]. Prefabricated zirconia crowns for primary teeth were introduced to the market over 10 years ago, and have been proven to have better results than other esthetic crowns in terms of esthetics, strength, and biocompatibility, such as gingival and periodontal health [11,12]. Recently, prefabricated zirconia crowns have been used for the treatment of deciduous teeth to provide a more durable and esthetic alternative. However, these crowns also have several drawbacks, such as high costs and wear on antagonist teeth [13]. Therefore, continuous efforts are required to develop esthetic restorative materials for primary teeth that can overcome the problems of preceding crowns and satisfy esthetics.

With the development of computer-aided design/computer-aided manufacturing (CAD/CAM), three-dimensional (3D) dental printing systems are emerging as remarkable technologies in the dental field. Digital processes in dental laboratories provide greater accuracy and reproducibility more quickly and easily at a lower cost. Three-dimensional printing technology is already being used in dentistry; for example, rapid prototyping skulls, computer-guided implant surgical templates, custom impression trays, and interim prostheses can be fabricated by 3D printing [14,15]. There are several studies on provisional prostheses, but few studies on final prostheses due to the absence of printable materials for definitive prostheses. In the same context, there are only a few studies using 3D printed resin crowns for restoring primary teeth [16,17]. As 3D printing technology and printable materials continue to advance, it is possible to fabricate long-lasting interim or final prostheses which can withstand both high stress and various chemical processes present in the oral cavity, while satisfying safety requirements [15,18,19].

Against this backdrop, this study aimed to determine whether 3D printing resins can be used for fixed prostheses of primary teeth by comparing the fracture resistance, biaxial flexural strength (BFS) and dynamic mechanical analysis (DMA) of two printable materials and prefabricated zirconia crowns. To determine the appropriate thickness for the 3D printed resin crown, this study was designed to identify the difference in strength according to various thicknesses. The null hypothesis of this study was that specimen thickness or types of printable resins do not affect fracture resistance and BFS.

## 2. Materials and Methods

### 2.1. Fracture Resistance

#### 2.1.1. Metal Die and Specimen Preparation

Two types of 3D printed resin crowns (Graphy TC-80DP [GP], Graphy Inc., Seoul, Korea; NextDent C&B MFH [NXT], NextDent B.V., Soesterberg, The Netherlands) and prefabricated zirconia crowns (NuSmile ZR [NS], NuSmile, Houston, TX, USA) were prepared (n = 15 per group). The specifications of the 3D printing resin materials are presented in Table 1.

A negative replica was fabricated with polyvinyl siloxane impression material (Examix NDS, GC Corporation, Tokyo, Japan) using NS zirconia crown, size number “4” crown of the primary mandibular second molar, and was allowed to set for 24 h. This impression was then scanned using an intraoral scanner (Cerec Omnicam, Sirona, Bensheim, Germany) and used to fabricate idealized Co-Cr dies by a milling procedure.

Three-dimensionally printed resin crowns were designed using Exocad software (Exocad Gmbh, Darmstadt, Germany) to have a uniform thickness on all surfaces, including occlusal, buccal, lingual, and proximal surfaces (Figure 1a). The GP and NXT samples were then printed using each company’s digital light processing (DLP) printers according to the layer thickness recommended by manufacturers (Table 2). After printing was complete, samples were washed for 5 min in an ultrasonic washing machine (Twin Tornado, Medifive, Seoul, Korea) with resin cleaner (Twin 3D Cleaner, Medifive, Seoul, Korea) to remove excessive resin monomers. The post-curing process for GP samples was conducted for 30 × 30 min in a post-curing unit (The CureM U102H, Graphy Inc., Seoul, Korea), and the NXT samples were post-cured for 30 min using a post-curing machine (LC-3DPrint box, NextDent, 3D systems, Rock Hill, SC, USA). Figure 1b shows the crowns used in this study. The thicknesses of 3D printed resin crowns were 0.4, 0.7, and 1.0 mm, within the error of ±0.05 mm. The thickness of NS crowns was also measured at the central point of all aspects including occlusal, buccal, lingual, mesial, and distal surfaces, within the error of ± 0.01 mm using the Iwanson spring measuring caliper (Hu-Friedy, Chicago, IL, USA).

All crowns and dies were tried on to ensure passive fit. Crowns were cemented onto the dies according to the manufacturer’s instructions with Scotchbond Universal Adhesive (3M ESPE, St. Paul, MN, USA) and resin cement (RelyX™ Ultimate, 3M ESPE, St. Paul, MN, USA). The die–crown units were then stored in distilled water at 37 °C for 24 h.

#### 2.1.2. Fracture Resistance Measurements

Each die–crown unit was placed in a universal testing machine (Instron 3366; Instron Co., Norwood, MA, USA) (Figure 2). The force was delivered through a stainless-steel ball fixture with a 7.5 mm diameter. A load was applied on the occlusal surface of the crowns with a cross-head speed of 1.0 mm/min until the crown fractured, and the force required to fracture it was recorded in Newton (N). Fractures were determined through audio or mechanical detection during loading.

### 2.2. Biaxial Flexural Strength (BFS)

#### 2.2.1. Three-Dimensional Printed Resin Disc Preparation

In total, 44 disc-shaped specimens with 12 mm diameter and 0.4, 0.7, and 1.0 mm thickness were fabricated by DLP printers using two types of 3D printing resin materials (GP and NXT). Each specimen was measured using a digital vernier caliper (Mitutoyo Corporation, Tokyo, Japan) within the error of ±0.01 mm. Prepared specimens were then immersed in distilled water at 37 °C for 24 h.

#### 2.2.2. Biaxial Flexural Strength Measurements

Specimens were tested on a universal testing machine (Instron 3366; Instron Co., Norwood, MA, USA) with piston-on-three-ball apparatus according to ISO 6872. Each specimen was positioned on three steel balls with a diameter of 2.5 mm, which were arranged in a circular shape with a diameter of 10 mm and separately arranged 120° apart from each other. The loading piston tip had a diameter of 1.4 mm, and a cross-head speed of 1.0 mm/min. The load was continuously applied until the specimens fractured. The load at fracture was recorded in N and then analyzed using Weibull analysis. The BFS was calculated as follows:σ = −0.2387P (X − Y)/d^2^
X = (1 + v)In(r_2_/r_3_)^2^ + ([1 − v]/2) (r_2_/r_3_)^2^
Y = (1 + v) (1 + In[r_1_/r_3_]^2^) + (1 − v)(r_1_/r_3_)^2^
where σ is the BFS (MPa), P is the fracture load (N), d is the disc specimen thickness (mm), v is the Poisson’s ratio (0.24), r_1_ is the radius of the support circle (5 mm), r_2_ is the radius of the loaded area (0.7 mm), and r_3_ is the radius of the specimen (6 mm).

### 2.3. Dynamic Mechanical Analysis (DMA)

Polymer materials have fundamentally different properties from metal and ceramic materials. In particular, there is a big difference in the rheological part [20,21,22]. It has an aliphatic carbon chain and is sensitive to temperature and strain because it is affected by electric charge interaction. Therefore, in this study, the experiment is conducted as a reference for the understanding of the overall behavior of photocurable resins and the long-term clinical environment, as well as the short-term clinical environment. It is correct to set the environment at 5–55 °C according to the ISO 10,477 standard, but the temperature range is set as follows in the evaluation of the original properties and long-term reliability of polymer materials [23]. To indirectly measure the material thermal dynamics of GP and NXT, DMA (Q800, TA Instruments, Wakefield, MA, USA) was performed for thermomechanical analysis between −30 and 120 °C with a frequency of 1 Hz and a strain rate of 0.1% using the dual cantilever method. Moreover, to analyze the viscoelastic properties of the material, the analysis was performed using a dual cantilever method at a frequency of 0.01–100 Hz with 0.1% strain at 37 °C. DMA samples with dimensions of 12.5 × 3 × 60 mm were printed.

### 2.4. Statistical Analyses

Statistical analyses were performed using statistical software (SPSS 25.0, IBM Corp., New York, NY, USA). Data were explored for normality by assessing the data distribution and using the Kolmogorov–Smirnov and Shapiro–Wilk tests. One-way analysis of variance (ANOVA) was used to analyze fracture resistance and BFS according to specimen thickness, and Student’s *t*-test was used to compare fracture resistance and BFS according to material. Tukey’s post hoc test was used for inter-group comparisons. The results were considered statistically significant at 95% confidence intervals (CI) and at a significance level of 0.05.

## 3. Results

### 3.1. Fracture Resistance

Table 3 and Figure 3 shows the mean fracture resistance of the GP, NXT, and NS groups. One-way ANOVA showed a statistically significant difference in fracture resistance according to crown thickness in GP (F = 6.215, *p* < 0.005) and NXT (F = 66.526, *p* < 0.001). Tukey’s post hoc test indicated that the mean value of the GP group was highest with 0.4 mm thickness and lowest with 0.7 mm. The NXT group showed the highest mean value with 1.0 mm thickness and the lowest mean value with 0.4 mm thickness.

The thicknesses of the NS crown, which were measured in this study on the number “4” crown of the primary mandibular second molar, were 0.69 mm at the central pit and 0.68, 0.67, 0.67, and 0.68 mm at the middle point of the buccal, lingual, mesial, and distal surfaces, respectively. The thickness of the NS crown was similar to those of the GP and NXT crowns (0.7 mm). Therefore, the NS group was compared with the 0.7-mm-thickness 3D printed resin crowns. Student’s *t*-test showed that the fracture resistance between the GP and NXT groups was statistically significant for both 0.4mm and 1.0 mm thicknesses (*p* < 0.001). However, for 0.7 mm thickness, one-way ANOVA showed no significant difference among the GP, NXT, and NS groups (*p* > 0.05) (Table 3).

### 3.2. Biaxial Flexural Strength (BFS)

Table 4 and Figure 4 show the mechanical properties of all test groups, including the mean and standard deviation of the BFS, the Weibull modulus (*m*), characteristic strength (*σ**_o_*), and Weibull distribution regression (R^2^). For all thicknesses, Student’s *t*-test showed that the mean value of GP was higher than that of NXT (*p* < 0.001). One-way ANOVA showed a statistically significant difference in BFS according to crown thickness in GP (F = 475.847, *p* < 0.001) and NXT (F = 131.061, *p* < 0.001). Tukey’s post hoc test indicated that the mean value of the GP and NXT groups was highest with 0.4 mm thickness and lowest with 1.0 mm thickness. In both the GP and NXT groups, specimen thickness and BFS were inversely proportional.

BFS was analyzed by Weibull to calculate the Weibull modulus, which represents the reliability of the material and the characteristic strength, which provides an estimated lifetime of the material, and 63.2% of the specimens failed at the characteristic strength. The obtained Weibull parameters were used to plot the failure probability line for each material with varied thicknesses, as shown in Figure 5. The plots describe the survival and failure times of the six test groups, and according to the linear fitting lines, under the same failure probability, GP material printed with 0.4 mm thickness withstand the maximum stress, and NXT 1.0 mm thickness withstand the least.

Additionally, the reliabilities (or survival probabilities) of the materials at different thicknesses were plotted with respect to the BFS in Figure 6. The specific survival probability values at 50, 150, and 250 MPa were calculated and are shown in Table 5 for the evaluation of their prostheses performances in the clinic. At 50 MPa, both materials showed similar reliabilities at all thicknesses. At 150 MPa, GP showed higher reliability than NXT at thicknesses 0.7 mm and 1.0 mm. Lastly, at 250 MPa, GP exhibited significantly higher reliabilities than NXT at 0.7 and 1.0 mm thicknesses.

Figure 7 shows the typical fracture patterns of GP and NXT after the BFS test. Most of the samples appeared to be broken into two or more pieces, whereas GP 0.4-mm-thick specimens were not completely separated and were still connected.

### 3.3. Dynamic Mechanical Analysis (DMA)

DMA was performed to measure the storage modulus according to the temperature and rheological behavior according to the frequency of photocurable, acrylic-based GP and NXT resins. The storage modulus curves of the cured GP and NXT resins are shown in Figure 8a,b. The temperature increased from −30 °C to 120 °C at a rate of 5 °C/min, and when it vibrated at 1 Hz, the storage modulus of GP and NXT decreased. The storage modulus of NXT was higher than that of GP. The decrease in the storage modulus with temperature was significantly better in NXT than in GP. The storage moduli at −30 °C were 2954 and 4403 MPa for GP and NXT, respectively. However, GP and NXT showed storage moduli of 58 and 533 MPa, respectively, in an atmosphere where the temperature was increased to 120 °C. Furthermore, in the case of the frequency sweep conducted at 37 °C, GP and NXT showed storage moduli of 2389 and 2695 MPa, respectively, at a frequency of 0.01 Hz. Moreover, at 100 Hz, GP showed a storage modulus of 2847 MPa and NXT of 3894 MPa.

The tan (δ) values representing the ratio of the loss factor to the storage factor for GP and NXT resins are shown in Figure 8e,f. At 37 °C, it did not show a significant change from 0.01 to 100 Hz and maintained a stable level. Therefore, as the temperature increased, the loss modulus of the material increased as shown in Figure 8c, and the tan (δ) value also increased. In this experiment, because measuring the Tg was not the goal, only the behavior according to the temperature rise was measured.

## 4. Discussion

With recent developments in digital dentistry, the use of 3D printing systems for fabricating esthetic prostheses has become favorable and reliable. For the past few decades, prefabricated zirconia crowns have been used to restore deciduous teeth esthetically [24]. However, the zirconia crowns currently available on the market are limited in size, and it is difficult to modify their forms. In particular, it is difficult to use them in patients with severe space loss due to large decay or patients with unique crown forms. Therefore, the application of 3D printing systems for restoring primary molars can be a new alternative for the management of primary dentition. Unlike other prefabricated crowns, such as stainless steel crowns or zirconia crowns, the 3D printed resin crown can be produced in a size and form optimized for each patient. Furthermore, since mass production using 3D printing technology is possible, there is an advantage in that various sizes of crowns can be prepared and used as prefabricated resin crowns. Thus, to investigate whether 3D-printed resin crowns can be used in children, this study aimed to determine the force required to fracture 3D printed resin crowns (in the primary molar areas) and to compare the mechanical properties of two types of printable photopolymer resins. Although in vitro studies cannot fully reproduce clinical conditions, they can provide directions for clinicians to decide on the use of the tested materials.

Based on the results of this study, 3D printed resin crowns had clinically comparable fracture resistance. Bite force increases with age from childhood onward. Braun et al. (1993) documented that the maximum bite force increased from 78 N at 6–8 years to 176 N at 18–20 years [25]. Owais et al. (2013) reported that the maximum bite force increased from 176 N in the early primary stage to 433 N in the late mixed stage [26]. The mean force values required to fracture 3D printed resin crowns in this study were over 1262.5 N in the NXT group and over 1491.6 N in the GP group. Therefore, all crowns with various thicknesses tested in this study were able to withstand the previously reported maximum bite force values in young children. Additionally, Chong et al. (2016) reported that the maximum occlusal forces in young and older adults were 541.4 and 420.5 N, respectively, which suggests that the fracture resistance of 3D printed resin crowns in this study exceeds the natural bite force produced by all age groups [27]. The findings of this study are similar to those of a study conducted by Al-Halabi et al. (2020), who reported that the fracture resistance of 3D printed resin crowns in the lower second primary molar was 1495.05 N [16].

Previous studies have shown that the fracture resistance of various prefabricated zirconia crowns is suitable for withstanding masticatory forces in pediatric patients [28,29]. Among them, the NS zirconia crown, most commonly used worldwide, was found to have 0.68 ± 0.02 mm thickness in primary second molar areas. Therefore, in this study, we decided to compare NS zirconia crown with 0.7-mm-thick 3D printed resin crowns, and found that there were no statistically significant differences in fracture resistance. Thus, compared with prefabricated zirconia crowns, 3D printed resin crowns were also found to have optimum fracture resistance.

Uniaxial strength tests, such as 3-point or 4-point bending tests, have long been used to determine the strengths of brittle materials. However, these uniaxial flexure testing methods can produce variations of results owing to defects or flaws within the edges of samples [30]. In contrast, using a multiaxial loading method, such as the BFS test, can prevent premature failures from flaws or cracks through three balls that contact evenly with warped specimens. Moreover, the multiaxial test can mimic the mastication process more than the uniaxial flexure test, which means that BFS can provide clinicians with more reliable data when selecting brittle restorative materials which can withstand chewing pressure [31,32]. Thus, the BFS test based on ISO 6872 was chosen in this study. BFS decreased as the thickness increased in both GP and NXT groups, and at the same thickness, the BFS of GP was significantly higher than that of NXT. This may be due to differences in the components of each 3D printed resin material. A photocurable material has the property of showing a low strain rate as it has a high cross-linking density. The cross-linking density changes depending on the number of functional groups in the acrylic resin constituting the material and the molecular weight of the oligomer. Low molecular weight, many acrylic functional groups, and nano- or sub-nano-sized additives result in high cross-linking density. When inorganic fillers or pigments are added, the filler prevents the deformation of the polymer resin matrix. This trend is more pronounced in thicker BSF test specimens. GP has strong mechanical strength because the composed oligomer has a high molecular weight. However, it is flexible because it has a low degree of cross-linking. This is the reason why the strength of the 0.4-mm-thick specimen was high during the BSF test. In a similar context, most of the specimens in this study were fractured into two or more pieces, whereas nine specimens of 0.4-mm-thick GP were not easily separated and were simply transformed even if they were already broken.

Regarding the characteristic strength, which represents the strength at which 63.2% of the specimens would fail, GP showed an average of 2.89 times higher characteristic strength values than NXT. In addition, the characteristic strengths increased with decreasing thickness for both materials. Regarding the Weibull modulus, which is used to determine the structural reliability of the material, no statistical differences were found between the two groups.

Among all experimental groups in this study, the lowest BFS was shown by 1.0-mm-thick NXT, which was 177.8 MPa. At 50 MPa, both GP and NXT showed more than 99.99% of survival probabilities, and at 150 MPa, both materials showed more than 90.6% survival probabilities. Thus, it was confirmed that GP and NXT would not fail at 50 MPa, the minimum flexural strength requirement for polymer-based crowns, which means that all specimens tested can endure the clinical flexural strength required. This result corresponds with the previous study that reported that all four types of 3D printed resins used for provisional dental restorations, Formlabs, Crowntec, Permanent Bridge Resin, and NextDent, showed high reliability at 50 MPa [33]. However, this minimum requirement of 50 MPa established at ISO 10,477 was based on a three-point bending test, and agreement on the mechanical evaluation of resin composites, other than ceramic, has yet to be reached. Therefore, comparisons between the uniaxial and biaxial flexural strength tests should be performed carefully.

As shown in Figure 8, the difference between GP and NXT materials was confirmed through DMA. At low frequencies, NXT showed a higher storage modulus than GP, and this phenomenon can be interpreted in connection with the fact that NXT showed low deformation tolerance even for the BFS test. NXT is expected to have a higher cross-linking density than GP. Alternatively, nano- and sub-nano-sized additives may have been used as components in NXT resin. On the other hand, GP showed higher stability than NXT with respect to temperature change. This is due to the oligomer which constitutes the GP resin that contains urethane acrylic resin which has a high molecular weight.

A limitation of the current study is that it was an in vitro study. In vitro studies cannot accurately reproduce the environment of the oral cavity, which exhibits various chemical and mechanical characteristics. Furthermore, unlike other prefabricated crowns, it is not yet known whether resin crowns can be used as prefabricated crowns. Therefore, further research should be conducted on other mechanical properties, such as fatigue strength, wear resistance, solubility, and permeability, and on whether adhesion strength or physical properties change over time after printing.

## 5. Conclusions

In this study, we evaluated the mechanical properties of two commercially available 3D printing resins and prefabricated zirconia crown for restoring primary teeth esthetically. There were significant differences in fracture resistance and BFS among the experimental groups according to specimen thickness and printable resins; however, all groups tested were demonstrated to have clinical applicability. Moreover, there was no significant difference in the strength of 3D printed resin crowns compared to that of prefabricated zirconia crowns. In addition, both GP and NXT groups were confirmed to be stable at the oral temperature through DMA, and the causes of differences in mechanical properties, including viscoelasticity of the materials, were analyzed by comparing the polymer behavior from a rheological point of view. Consequently, 3D printed resin crowns could be a new alternative to restoring primary molars while satisfying the need for esthetics.

## Figures and Tables

**Figure 1 children-09-01445-f001:**
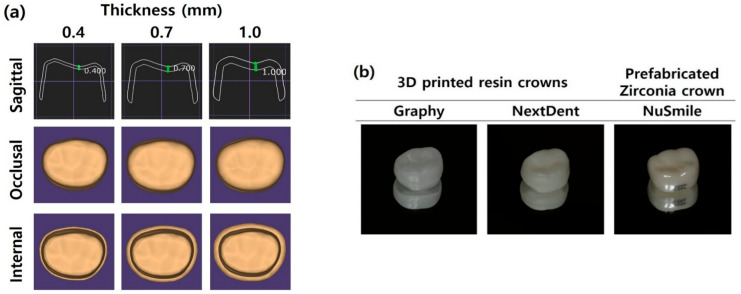
The shape of the crowns used in this study: (**a**) CAD/CAM designs of 3D printed resin crowns according to thickness and (**b**) prepared 3D printed resin crowns and prefabricated zirconia crown.

**Figure 2 children-09-01445-f002:**
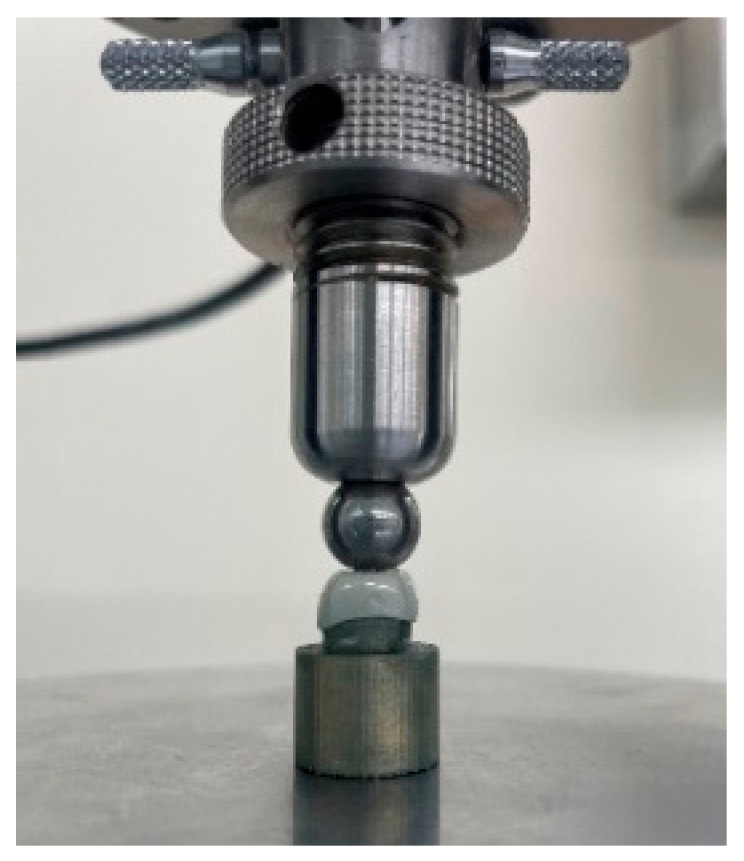
Die–crown unit positioned in the universal testing machine.

**Figure 3 children-09-01445-f003:**
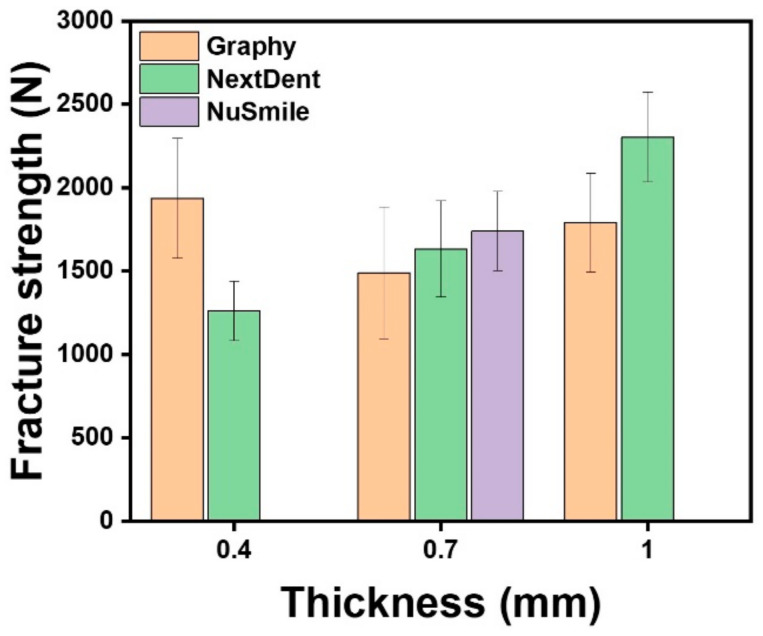
Mean fracture resistance of the various experimental groups according to thickness.

**Figure 4 children-09-01445-f004:**
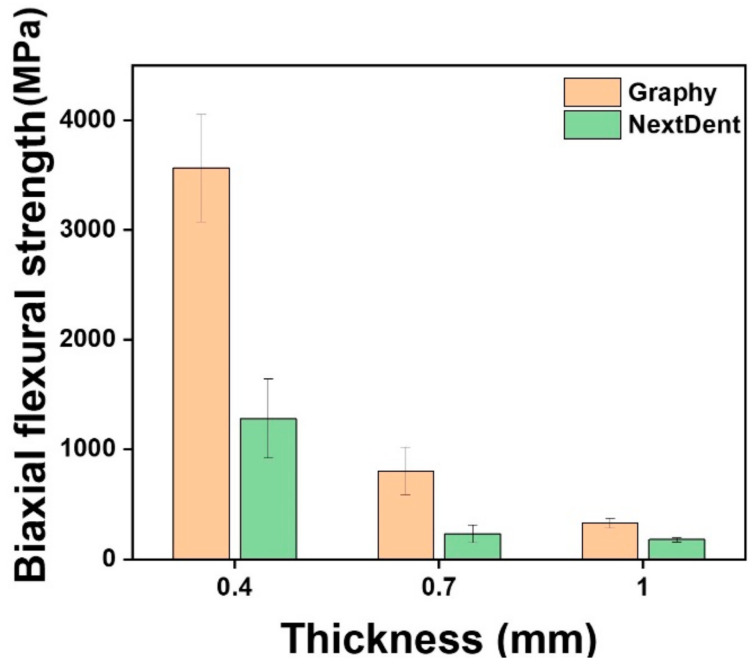
Mean biaxial flexural strength (MPa) of the various experimental groups.

**Figure 5 children-09-01445-f005:**
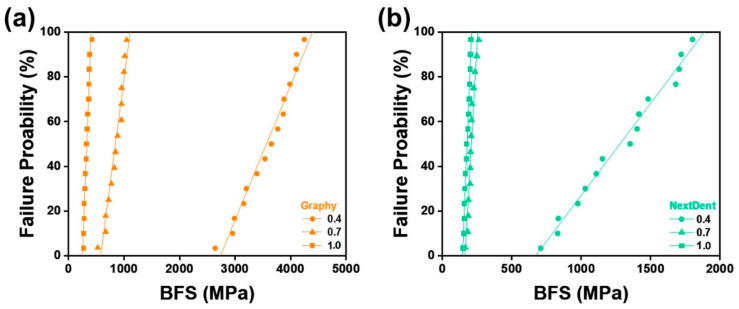
Failure probability according to Weibull analysis of biaxial flexural strength (BFS): (**a**) Graphy, (**b**) NextDent.

**Figure 6 children-09-01445-f006:**
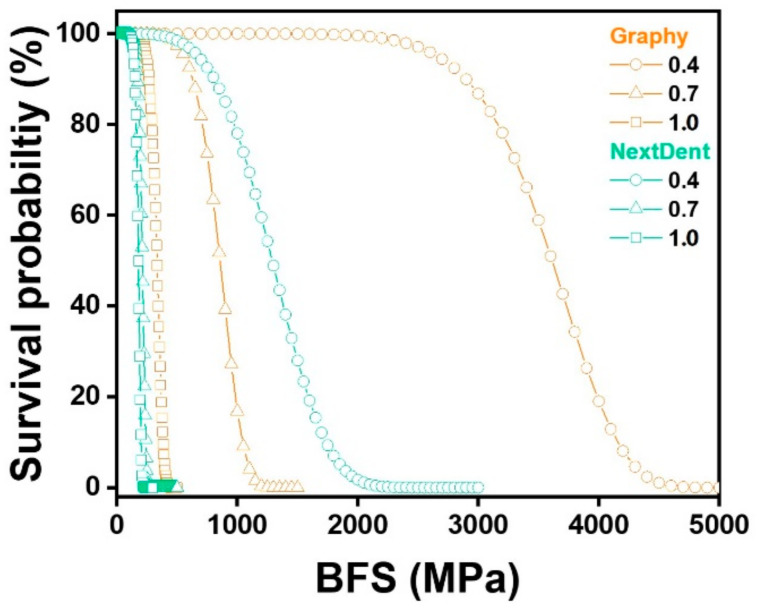
Weibull survival probability based on biaxial flexural strength (BFS) for thicknesses 0.4, 0.7, and 1.0 mm for Graphy and NextDent materials.

**Figure 7 children-09-01445-f007:**
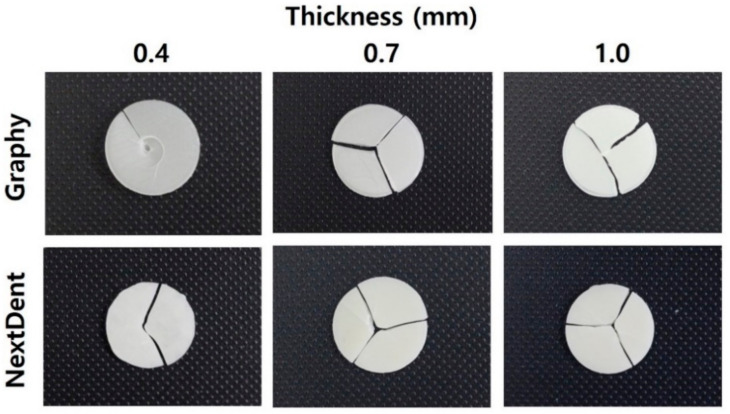
Representative fractured patterns of the various experimental groups.

**Figure 8 children-09-01445-f008:**
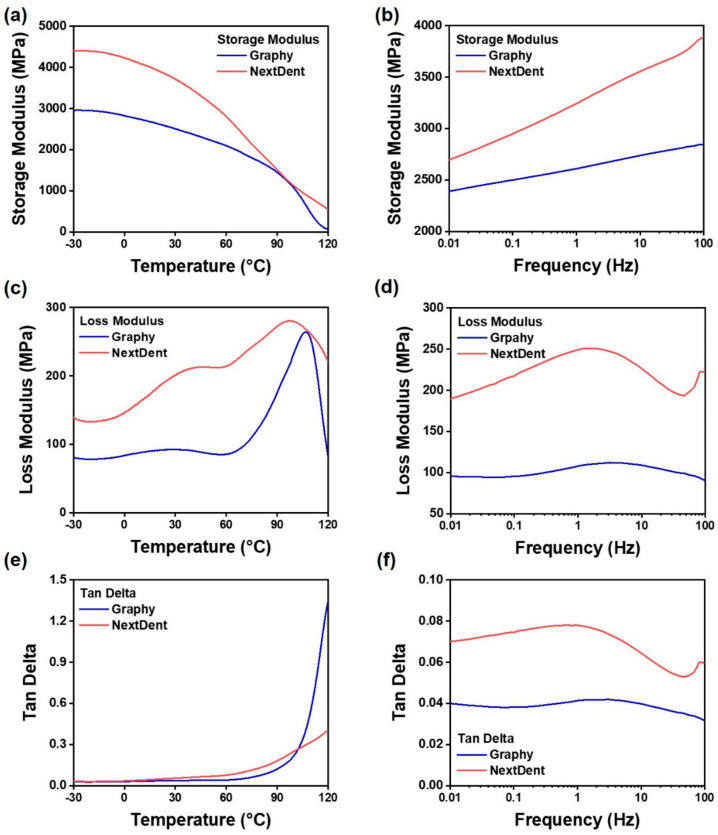
Illustration of dynamic mechanical property measurement results using temperature sweep and frequency sweep mode using DMA for 3D printing resins. In cases (**a**–**f**), measurements were performed using the dual cantilever mode of DMA, and the specimen size was output based on TA’s reference steel sample. (**a**,**c**,**e**) The range of −30 to 120 °C was measured under the conditions of 1 Hz and 0.1% using the temperature sweep mode of DMA. (**b**,**d**,**f**) The range of 0.01 to 100 Hz was measured under the conditions of 1 Hz and 0.1% using the frequency sweep mode of DMA. (**a**,**b**) The storage modulus of the materials is a graph. (**c**,**d**) Figures analyzing the loss modulus. (**e**,**f**) Graphs of the loss/storage modulus ratio (Tan delta) of the materials.

**Table 1 children-09-01445-t001:** Material composition of 3D printing materials used in this study provided by the manufacturers.

Manufacturer	Product	Shade	Composition	Batch Number
Graphy (GP)	TC-80DP	A1	Methacrylate oligomer based on polyurethane resin, phosphine oxides, pigment	1-B1220K11-003
NextDent (NXT)	C&B MFH	N1	>90% methacrylic oligomers, methacrylate monomer, <3% phosphine oxides, pigment	WX151N01

**Table 2 children-09-01445-t002:** Characteristics of 3D printing devices.

Printing Materials	Printer	Manufacturer	Printing Volume	Layer Thickness	Wavelength	Pixel Pitch
GP	Sprintray Pro95	Graphy Inc.	192 × 100 × 200 mm	50–100 μm	405 nm	95 µm
NXT	NextDent 5100	NextDent B.V.	124.8 × 70.2 × 196 mm	30–100 μm	405 nm	65 µm

**Table 3 children-09-01445-t003:** The mean fracture resistance according to the thickness of crowns.

Thickness (mm)	Materials	Force Required for Fracture (N)						
		N	Mean	SD	Min	Max	95% CI	*p* Value
0.4	GP	15	1937.4	360.6	1326.0	2621.0	1737.7–2137.1	0.000 *
	NXT	15	1262.5	178.6	912.2	1526.4	1163.6–1361.4	
0.7	GP	15	1491.6	394.6	924.0	2197.2	1273.1–1710.2	0.103
	NXT	15	1634.4	289.3	1200.8	2025.8	1474.2–1794.7	
	NS	15	1742.3	237.5	1296.9	2151.0	1610.8–1873.9	
1.0	GP	15	1792.2	297.5	1279.1	2163.9	1627.4–1956.9	0.000 *
	NXT	15	2303.7	269.6	1848.2	2657.1	2154.4–2453.0	

* Statistically significant at *p* ≤ 0.05.

**Table 4 children-09-01445-t004:** Weibull analysis of biaxial flexural strength.

Thickness (mm)	Materials	N	Biaxial Flexural Strength (MPa)			*m*	*σ_o_*	R^2^
			Mean	SD	*p* Value			
0.4	GP	15	3564.6	489.1	0.000 *	8.55	3770.3	0.97
	NXT	15	1279.9	359.9		4.03	1412.7	0.95
0.7	GP	14	845.4	155.3	0.000 *	6.12	910.07	0.98
	NXT	14	209.3	26.58		9.47	220.35	0.92
1.0	GP	15	329.3	45.40	0.000 *	8.55	348.48	0.89
	NXT	15	177.8	19.46		10.7	186.23	0.92

* Statistically significant at *p* ≤ 0.05; N = number of samples; SD = standard deviation; *m* = Weibull modulus; *σ_o_* = characteristic strength; R^2^ = Weibull distribution regression.

**Table 5 children-09-01445-t005:** Reliability of Graphy and NextDent materials at different thicknesses under compressive forces of 50, 150, and 250 MPa.

Thickness (mm)	Materials	50 MPa	150 MPa	250 MPa
0.4	GP	100.0000	100.0000	100.0000
	NXT	99.9999	99.9999	99.9999
0.7	GP	100.0000	99.9984	99.9632
	NXT	99.9999	97.4110	3.6763
1.0	GP	100.0000	99.9259	94.3227
	NXT	99.9999	90.6396	0.0000

## Data Availability

Data will be made available upon reasonable request to the authors.

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
