# Peer review of "Novel 3D Printed Resin Crowns for Primary Molars: In Vitro Study of Fracture Resistance, Biaxial Flexural Strength, and Dynamic Mechanical Analysis"

_children, 2022, doi:10.3390/children9101445_

Round 1

Reviewer 1 Report

I enjoyed reading this research. The experimental design was well planned and executed.

Author Response

Thank you for your interest in our research and for your advice.

As your suggestion, we revised and corrected the overall English with great care. Thank you. 

Reviewer 2 Report

well written manuscript

Author Response

Thank you for your interest in our research and for your advice.

Reviewer 3 Report

Language needs improvement, especially in the Introduction and the Discussion. One of the major concerns here is the English. The authors should have the manuscript looked at for language and sentence composition. There are a lot of sentences in the manuscript that do not make sense because of the English. 

Material and method: The Batch Number of the materials used does not appear, please add it.

Biocompatible printed materials carry a very specific series of post-processing, in this article it does not appear, neither what printer has been used, nor how they have been polymerized or cleaned, I think it is something essential, since an incorrect handling of the process would affect its mechanical properties . It is essential that this appears. Perhaps you could cite this article that talks about that topic.  Guerrero-Gironés J, López-García S, Pecci-Lloret MR, Pecci-Lloret MP, Rodríguez Lozano FJ, García-Bernal D. In vitro biocompatibility testing of 3D printing and conventional resins for occlusal devices. J Dent. 2022 Aug;123:104163. doi: 10.1016/j.jdent.2022.104163. Epub 2022 May 14. PMID: 35577252.

Discussion.

What is the importance of this study? Do you think this study is similar to others already carried out? What is new about this study?

Something to keep in mind is the clinical importance. To print a pediatric crown we need to design it, print it, clean it and polymerize it. So at least 40 minutes would go away, while metal or zirconia crowns are put on immediately. Does it have any clinical interest to place printed crowns in our pediatric patients in whom we need immediacy?

Conclusions were not totally supported by the data showed. 

Author Response

Thank you for your interest in our research and for your advice.

We revised the manuscript with great care, so we would appreciate it if you read and reconsider it again. 

The response is attached below.

Thank you.

Reviewer 4 Report

At Dynamic Mechanical Analysis: Justify why you used those conditions of analysis in the methodology, and in the discussion what is their clinical impact.}

In the methods section,  2.2.3.1 is the wrong sequence.

in the paragraph "Statistical analyses were performed using statistical software (SPSS 25.0, IBM Corp., New York, NY, USA). To analyze fracture resistance, Student t test and one-way analysis of variance (ANOVA) were used. Two-way ANOVA was used to analyze the BFS. The results were considered statistically significant at 95% confidence intervals (CI) and at a significance level of 0.05. Tukey’s post hoc test was used for inter-group comparisons. Data were explored for normality by assessing the data distribution and using Kolmogorov–Smirnov and Shapiro–Wilk tests".   First, the data normality analysis should have been described, specifying the number of samples and then explaining the inferential statistical tests. 

Why were both t-test and ANOVA used for fracture toughness? this is not suitable.

Why was one-way ANOVA used in fracture strength and two-way ANOVA in BFS? Justify the statistical tests according to the hypotheses arising for each variable. Please, write the hypotheses after the objective of the investigation in the introductory section.

The description of the results is repetitive with the information in the table. Please summarize to the most important. 

The plots of the dynamic mechanical properties seem inappropriate in the range of -30 to 120 °C. The temperature must be consistent with some justification for the use of the polymer or its manufacture, and/or storage. A temperature as extreme as -30 does not contribute to the research work. Please, adjust the intervals in the graphs presented.

The discussion section is appropriate and current. Although the conclusions are very extensive and repeat what is already in the text of the manuscript, I suggest reducing this section.

Author Response

(The authors gave the same response as above.)

Round 2

Reviewer 3 Report

The article has improved qualitatively and quantitatively in general.